# CHAMELEON: LEARNING MODEL INITIALIZATIONS ACROSS TASKS WITH DIFFERENT SCHEMAS

## ABSTRACT

Parametric models, and particularly neural networks, require weight initialization as a starting point for gradient-based optimization. Recent work shows that an initial parameter set can be learned from a population of supervised learning tasks that enables a fast convergence for unseen tasks even when only a handful of instances is available (model-agnostic meta-learning). Currently, methods for learning model initializations are limited to a population of tasks sharing the same schema, i.e., the same number, order, type, and semantics of predictor and target variables. In this paper, we address the problem of meta-learning weight initialization across tasks with different schemas, for example, if the number of predictors varies across tasks, while they still share some variables. We propose Chameleon, a model that learns to align different predictor schemas to a common representation. In experiments on 23 datasets of the OpenML-CC18 benchmark, we show that Chameleon can successfully learn parameter initializations across tasks with different schemas, presenting, to the best of our knowledge, the first cross-dataset few-shot classification approach for unstructured data.

## 1 INTRODUCTION

Humans require only a few examples to correctly classify new instances of previously unknown objects. For example, it is sufficient to see a handful of images of a specific type of dog before being able to classify dogs of this type consistently. In contrast, deep learning models optimized in a classical supervised setup usually require a vast number of training examples to match human performance. A striking difference is that a human has already learned to classify countless other objects, while parameters of a neural network are typically initialized randomly. Previous approaches improved this starting point for gradient-based optimization by choosing a more robust random initialization (He et al., 2015) or by starting from a pretrained network (Pan & Yang, 2010). Still, models do not learn from only a handful of training examples even when applying these techniques. Moreover, established hyperparameter optimization methods (Schilling et al., 2016) are not capable of optimizing the model initialization due to the high-dimensional parameter space. Few-shot classification aims at correctly classifying unseen instances of a novel task with only a few labeled training instances given. This is typically accomplished by meta-learning across a set of training tasks, which consist of training and validation examples with given labels for a set of classes. The field has gained immense popularity among researchers after recent meta-learning approaches have shown that it is possible to learn a weight initialization across different tasks, which facilitates a faster convergence speed and thus enables classifying novel classes after seeing only a few instances (Finn et al., 2018). However, training a single model across different tasks is only feasible if all tasks share the same schema, meaning that all instances share one set of features in identical order. For that reason, most approaches demonstrate their performance on image data, which can be easily scaled to a fixed shape, whereas transforming unstructured data to a uniform schema is not trivial.

We want to extend popular approaches to operate invariant of schema, i.e., independent of order and shape, making it possible to use meta-learning approaches on unstructured data with varying feature spaces, e.g., learning a model from heart disease data that can accurately classify a few-shot task for diabetes detection that relies on similar features. Thus, we require a schema-invariant encoder that maps heart disease and diabetes data to one feature representation, which then can be used to train a single model via popular meta-learning algorithms like REPTILE (Nichol et al., 2018b).

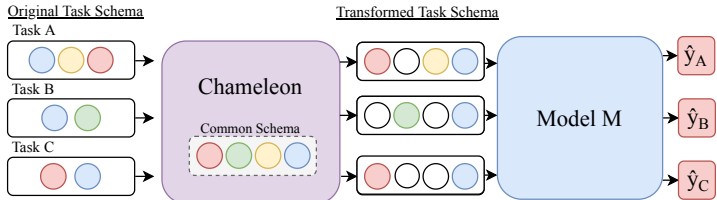

Figure 1: **Chameleon Pipeline**: Chameleon aims to encode tasks with different schemas to a shared representation with an uniform feature space, which can then be processed by any classifier. The left block represents tasks of the same domain with different schemas. The middle represents the aligned features in a fixed schema.

We propose a set-wise feature transformation model called CHAMELEON, named after a REPTILE capable of adjusting its colors according to the environment in which it is located. CHAMELEON projects different schemas to a fixed input space while keeping features from different tasks but of the same type or distribution in the same position, as illustrated by Figure 1. Our model learns to compute a task-specific reordering matrix that, when multiplied with the original input, aligns the schema of unstructured tasks to a common representation while behaving invariant to the order of input features.

Our main contributions are as follows: (1) We show how our proposed method CHAMELEON can learn to align varying feature spaces to a common representation. (2) We propose the first approach to tackle few-shot classification for tasks with different schemas. (3) In experiments on 23 datasets of the OpenML-CC18 benchmark (Bischl et al., 2017) collection, we demonstrate how current meta-learning approaches can successfully learn a model initialization across tasks with different schemas as long as they share some variables with respect to their type or semantics. (4) Although an alignment makes little sense to be performed on top of structured data such as images which can be easily rescaled, we demonstrate how CHAMELEON can align latent embeddings of two image datasets generated with different neural networks.

## 2 RELATED WORK

Our goal is to extend recent few-shot classification approaches that make use of optimization-based meta-learning by adding a feature alignment component that casts different inputs to a common schema, presenting the first approach working across tasks with different schema. In this section, we will discuss various works related to our approach.

Research on transfer learning (Pan & Yang, 2010; Sung et al., 2018; Gligic et al., 2020) has shown that training a model on different auxiliary tasks before actually fitting it to the target problem can provide better results if training data is scarce. Motivated by this, few-shot learning approaches try to generalize to novel tasks with unseen classes given only a few instances by first meta-learning across a set of training tasks (Duan et al., 2017; Finn et al., 2017b; Snell et al., 2017). A task $\tau$ consists of predictor data $X_\tau$, a target $Y_\tau$, a predefined training/test split $\tau = (X_\tau^{\text{train}}, Y_\tau^{\text{train}}, X_\tau^{\text{test}}, Y_\tau^{\text{test}})$ and a loss function $\mathcal{L}_\tau$. Typically, an $N$-way $K$-shot problem refers to a few-shot learning problem where each task consists of $N$ classes with $K$ training samples per class.

Heterogeneous Transfer Learning tries to tackle a similar problem setting as described in this work. In contrast to regular Transfer Learning, the feature spaces of the auxiliary tasks and the actual task differ and are often non-overlapping (Day & Khoshgoftaar, 2017). Many approaches require co-occurence data i.e. instances that can be found in both datasets (Wu et al., 2019; Qi et al., 2011), rely on jointly optimizing separate models for each dataset to propagate information (Zhao & Hoi, 2010; Yan et al., 2016), or utilize meta-features (Feuz & Cook, 2015). Oftentimes, these approaches operate on structured data e.g. images and text with different data distributions for the tasks at hand (Li et al., 2019; He et al., 2019). These datasets can thus be embedded in a shared space with standard models such as convolutional neural networks and transformer-based language models. However, none of these approaches are capable of training a single encoder that operates across a meta-dataset of tasks with different schema for unstructured data.

Early approaches like (Fe-Fei et al., 2003) already investigated the few-shot learning setting by representing prior knowledge as a probability density function. In recent years, various works proposed new model-based meta-learning approaches which rapidly improved the state-of-the-art few-shot learning benchmarks. Most prominently, this includes methods which rely on learning an embedding space for non-parametric metric approaches during inference time (Vinyals et al., 2016; Snell et al., 2017), and approaches which utilize an external memory which stores information about previously seen classes (Santoro et al., 2016; Munkhdalai & Yu, 2017). Several more recent meta-learning approaches have been developed which introduce architectures and parameterization techniques specifically suited for few-shot classification (Mishra et al., 2018; Shi et al., 2019; Wang & Chen, 2020) while others try to extract useful meta-features from datasets to improve hyper-parameter optimization (Jomaa et al., 2019).

In contrast, Finn et al. (2017a) showed that an optimization-based approach, which solely adapts the learning paradigm can be sufficient for learning across tasks. Model Agnostic Meta-Learning (MAML) describes a model initialization algorithm that is capable of training an arbitrary model $f$ across different tasks. Instead of sequentially training the model one task at a time, it uses update steps from different tasks to find a common gradient direction that achieves a fast convergence. In other words, for each meta-learning update, we would need an initial value for the model parameters $\theta$. Then, we sample a batch of tasks $\mathcal{T}$, and for each task $\tau \in \mathcal{T}$ we find an updated version of $\theta$ using $N$ examples from the task by performing gradient descent with learning rate $\alpha$ as in: $\theta'_\tau \leftarrow \theta - \alpha \nabla_\theta \mathcal{L}_\tau(f_\theta)$. The final update of $\theta$ with step size $\beta$ will be:

$$\theta \leftarrow \theta - \beta \frac{1}{|\mathcal{T}|} \nabla_\theta \sum_\tau \mathcal{L}_\tau(f_{\theta'_\tau}) \tag{1}$$

Finn et al. (2017a) state that MAML does not require learning an update rule (Ravi & Larochelle, 2016), or restricting their model architecture (Santoro et al., 2016). They extended their approach by incorporating a probabilistic component such that for a new task, the model is sampled from a distribution of models to guarantee a higher model diversification for ambiguous tasks (Finn et al., 2018). However, MAML requires to compute second-order derivatives, resulting in a computationally heavy approach. Nichol et al. (2018b) extend upon the first-order approximation given as an ablation by Finn et al. (2018), which numerically approximates Equation (1) by replacing the second derivative with the weights difference, s.t. the update rule used in REPTILE is given by:

$$\theta \leftarrow \theta - \beta \frac{1}{|\mathcal{T}|} \sum_\tau (\theta'_\tau - \theta) \tag{2}$$

which means we can use the difference between the previous and updated version as an approximation of the second-order derivatives to reduce computational cost. The serial version is presented in Algorithm (1).[1] All of these approaches rely on a fixed schema, i.e. the same set of features with identical alignment across all tasks. However, many similar datasets only share a subset of their features, while oftentimes having a different order or representation e.g. latent embeddings for two different image datasets generated by training two similar architectures. Most current few-shot classification approaches sample tasks from a single dataset by selecting a random subset of classes; although it is possible to train a single meta-model on two different image datasets as shown by Munkhdalai & Yu (2017) and Tseng et al. (2020) since the images can be scaled to a fixed size. Further research demonstrates that it is possible to learn a single model across different output sizes (Drumond et al., 2020). Recently, a meta-dataset for few-shot classification of image tasks was also published to promote meta-learning across multiple datasets (Triantafillou et al., 2020). Optimizing a single model across various datasets requires a shared feature space. Thus, it is required to align the features which is achieved by simply rescaling all instances in the case of image data which is not trivial for unstructured data. Recent work relies on preprocessing images to a one-dimensional latent embedding with an additional deep neural network. The authors Rusu et al. (2019) train a Wide Residual Network (Zagoruyko & Komodakis, 2016) on the meta-training data of MiniImageNet (Vinyals et al., 2016) to compute latent embeddings of the data which are then used for few-shot classification, demonstrating state-of-the-art results.

Finding a suitable initialization for deep network has long been a focus of machine learning research. Especially the initialization of Glorot & Bengio (2010) and later He et al. (2015) which emphasize

---

[1] Note that REPTILE does not require validation instances during meta-learning.

the importance of a scaled variance that depends on the layer inputs are widely used. Similar findings are also reported by Cao et al. (2019). Recently, Dauphin & Schoenholz (2019) showed that it is possible to learn a suitable initialization by optimizing the norms of the respective weights. So far, none of these methods tried to learn a common initialization across tasks with different schema.

We propose a novel feature alignment component named CHAMELEON, which enables state-of-the-art methods to learn how to work on top of tasks whose feature vector differ not only in their length but also their concrete alignment. Our model shares resemblance with scaled dot-product attention popularized by (Vaswani et al., 2017):

$$Attention(Q, K, V) = softmax(\frac{QK^T}{\sqrt{d_K}})V \tag{3}$$

where $Q$, $K$ and $V$ are matrices describing queries, keys and values, and $d_K$ is the dimensionality of the keys such that the softmax computes an attention mask which is then multiplied with the values $V$. In contrast to this, we pretrain the parametrized model CHAMELEON to compute a soft permutation matrix which can realign features across tasks with varying schema when multiplied with $V$ instead of computing a simple attention mask.

---

**Algorithm 1** REPTILE Nichol et al. (2018b)

---

**Input**: Meta-dataset $\mathcal{T} = \{(X_1, Y_1, \mathcal{L}_1), ..., (X_{|\mathcal{T}|}, Y_{|\mathcal{T}|}, \mathcal{L}_{|\mathcal{T}|})\}$, learning rate $\beta$
 1: Randomly initialize parameters $\theta$ of model $f$
 2: **for** iteration = 1, 2, ... **do**
 3:    Sample task $(X_\tau, Y_\tau, \mathcal{L}_\tau) \sim \mathcal{T}$
 4:    $\theta' \leftarrow \theta$
 5:    **for** k steps = 1,2,... **do**
 6:       $\theta' \leftarrow \theta' - \alpha\nabla_{\theta'}\mathcal{L}_\tau(Y_\tau, f(X_\tau; \theta'))$
 7:    **end for**
 8:    $\theta \leftarrow \theta - \beta(\theta' - \theta)$
 9: **end for**
10: **return** parameters $\theta$ of model $f$

---

## 3 METHODOLOGY

### 3.1 PROBLEM SETTING

We describe a classification dataset with vector-shaped predictors (i.e., no images, time series etc.) by a pair $(X, Y) \in \mathbb{R}^{N \times F} \times \{0, ..., C\}^N$, with predictors $X$ and targets $Y$, where $N$ denotes the number of instances, $F$ the number of predictors and $C$ the number of classes. Let $\mathcal{D}_F := \bigcup_{N \in \mathbb{N}} \mathbb{R}^{N \times F} \times \{0, ..., C\}^N$ be the space of all such datasets with $F$ predictors and $\mathcal{D} := \bigcup_{F \in \mathbb{N}} \mathcal{D}_F$ be the space of any such dataset. Let us also denote the space of all predictor matrices with $F$ predictors by $\mathcal{X}_F := \bigcup_{N \in \mathbb{N}} \mathbb{R}^{N \times F}$ and all predictor matrices by $\mathcal{X} := \bigcup_{F \in \mathbb{N}} \mathcal{X}_F$. Then a dataset $\tau = (X, Y) \in \mathcal{D}$ equipped with a predefined training/test split, i.e. the quadruplet $\tau = (X_\tau^{\text{train}}, Y_\tau^{\text{train}}, X_\tau^{\text{test}}, Y_\tau^{\text{test}})$ is called a *task*. A collection of such tasks $\mathcal{T} \subset \mathcal{D}$ is called a *meta-dataset*. Similar to splitting a single data set into a training and test part, one can split a meta-dataset $\mathcal{T} = \mathcal{T}^{\text{train}} \dot\cup \mathcal{T}^{\text{test}}$. The *schema* of a task $\tau$ then describes not only the number and order, but also the semantics of predictor variables $\{p_{\tau_1}, p_{\tau_2}, \ldots, p_{\tau_F}\}$ in $X_\tau^{train}$.

Consider a meta-dataset of correlated tasks $\mathcal{T} \subset \mathcal{D}$, such that the predictor variables $\{p_{\tau_1}, p_{\tau_2}, \ldots, p_{\tau_F}\}$ of any individual task $\tau$ are contained in a common set of predictor variables $\{p_1, p_2, \ldots, p_K\}$. Methods like REPTILE and MAML try to find the best initialization for a specific model, in this work referred to as $\hat{y}$, to operate on a set $\mathcal{T}$ of similar tasks. However, every task $\tau$ has to share the same schema of common size $K$, where similar features shared across tasks are in the same position. A feature-order invariant encoder is needed to map the data representation $X_\tau$ of tasks with varying input schema and feature length $F_\tau$ to a shared latent representation $\widetilde{X}_\tau$ with fixed feature length $K$:

$$\text{enc} \colon \mathcal{X} \longrightarrow \mathcal{X}_K, \ X_\tau \in \mathbb{R}^{N \times F_\tau} \longmapsto \widetilde{X}_\tau \in \mathbb{R}^{N \times K} \tag{4}$$

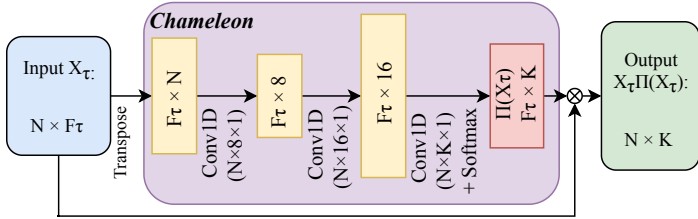

Figure 2: **The Chameleon Architecture:** $N$ represents the number of samples in $\tau$, $F_\tau$ is the number of features in $\tau$, and K is the number of features in the desired feature space. "Conv($a \times b \times c$)" is a convolution operation with $a$ input channels, filter size of $b$ and kernel length $c$.

where $N$ represents the number of instances in $X_\tau$, $F_\tau$ is the number of features of task $\tau$ which varies across tasks, and $K$ is the size of the desired feature space. By combining this encoder with model $\hat{y}$ that works on a fixed input size $K$ and outputs the predicted target e.g. binary classification, it is possible to apply the REPTILE algorithm to learn an initialization $\theta^{\text{init}}$ across tasks with different schema. The optimization objective then becomes the meta-loss for the combined network $f = \hat{y} \circ \text{enc}$ over a set of tasks $\mathcal{T}$:

$$\underset{\theta^{\text{init}}}{\text{argmin}} \quad \mathbb{E}_{\tau \sim \mathcal{T}} L_\tau \Big( Y_\tau^{\text{test}}, \ f\big(X_\tau^{\text{test}}; \theta_\tau^{(u)}\big)\Big) \quad \text{s.t.} \quad \theta_\tau^{(u)} = \mathcal{A}^{(u)}\Big(X_\tau^{\text{train}}, Y_\tau^{\text{train}}, L_\tau, f; \theta^{\text{init}}\Big) \quad (5)$$

where $\theta^{\text{init}}$ is the set of initial weights for the combined network $f$ consisting of enc with parameters $\theta_{\text{enc}}$ and model $\hat{y}$ with parameters $\theta_{\hat{y}}$, and $\theta_\tau^{(u)}$ are the updated weights after applying the learning procedure $\mathcal{A}$ for $u$ iterations on the task $\tau$ as defined in Algorithm 1 for the inner updates of REPTILE. It is important to mention that learning one weight parameterization across any heterogeneous set of tasks is extremely difficult since it is most likely impossible to find one initialization for two tasks with a vastly different number and type of features. By contrast, if two tasks share similar features, one can align the similar features to a common representation so that a model can directly learn across different tasks by transforming the tasks as illustrated in Figure 1.

## 3.2 CHAMELEON

Consider a set of tasks where a right stochastic matrix $\Pi_\tau$ exists for each task that reorders predictor data $X_\tau$ into $\widetilde{X}_\tau$ having the same schema for every task $\tau \in \mathcal{T}$:

$$\widetilde{X}_\tau = X_\tau \cdot \Pi_\tau, \text{where} \quad (6)$$

$$\underbrace{\begin{bmatrix} \tilde{x}_{1,1} & \cdots & \tilde{x}_{1,K} \\ \vdots & \ddots & \vdots \\ \tilde{x}_{N,1} & \cdots & \tilde{x}_{N,K} \end{bmatrix}}_{\widetilde{X}_\tau} = \underbrace{\begin{bmatrix} x_{1,1} & \cdots & x_{1,F_\tau} \\ \vdots & \ddots & \vdots \\ x_{N,1} & \cdots & x_{N,F_\tau} \end{bmatrix}}_{X_\tau} \cdot \underbrace{\begin{bmatrix} \pi_{1,1} & \cdots & \pi_{1,K} \\ \vdots & \ddots & \vdots \\ \pi_{F_\tau,1} & \cdots & \pi_{F_\tau,K} \end{bmatrix}}_{\Pi_\tau}$$

Every $x_{m,n}$ represents the feature $n$ of sample $m$. Every $\pi_{m,n}$ represent how much of feature $m$ (from samples in $X_\tau$) should be shifted to position $n$ in the adapted input $\widetilde{X}_\tau$. Finally, every $\tilde{x}_{m,n}$ represent the new feature $n$ of sample $m$ in $X_\tau$ with the adpated shape and size. In order to achieve the same $\widetilde{X}_\tau$ when permuting two features of a task $X_\tau$, we must simply permute the corresponding rows in $\Pi_\tau$ to achieve the same $\widetilde{X}_\tau$. Since $\Pi_\tau$ is a right stochastic matrix, the summation for every row of $\Pi_\tau$ is set to be equal to 1 as in $\sum_i \pi_{j,i} = 1$, so that each value in $\Pi_\tau$ simply states how much a feature is shifted to a corresponding position. For example: Consider that task $a$ has features [*apples, bananas, melons*] and task $b$ features [*lemons, bananas, apples*]. Both can be transformed to the same representation [*apples, lemons, bananas, melons*] by replacing missing features with zeros and reordering them. This transformation must have the same result for $a$ and $b$ independent of their feature order. In a real life scenario, features might come with different names or sometimes their similarity is not clear to the human eye. Note that a classic autoencoder is not capable of this as it is not invariant to the order of the features. Our proposed component, denoted by $\Phi$, takes a task as

input and outputs the corresponding reordering matrix:

$$\Phi(X_\tau, \theta_{\text{enc}}) = \hat{\Pi}_\tau \tag{7}$$

The function $\Phi$ is a neural network parameterized by $\theta_{\text{enc}}$. It consists of three 1D-convolutions, where the last one is the output layer that estimates the alignment matrix via a softmax activation. The input is first transposed to size $[F_\tau \times N]$ (where N is the number of samples) i.e., each feature is represented by a vector of instances. Each convolution has kernel length 1 (as the order of instances is arbitrary and thus needs to be permutation invariant) and a channel output size of 8, 16, and lastly $K$. The result is a reordering matrix displaying the relation of every original feature to each of the $K$ features in the target space. Each of these vectors passes through a softmax layer, computing the ratio of features in $X_\tau$ shifted to each position of $\widetilde{X}_\tau$. Finally, the reordering matrix can be multiplied with the input to compute the aligned task as defined in Equation (6). By using a kernel length of 1 in combination with the final matrix multiplication, the full architecture becomes permutation invariant in the feature dimension. Column-wise permuting the features of an input task leads to the corresponding row-wise permutation of the reordering matrix. Thus, multiplying both matrices results in the same aligned output independent of permutation. The overall architecture can be seen in Figure 2. The encoder necessary for training across tasks with different predictor vectors with REPTILE by optimizing Equation (5) is then given as:

$$\text{enc}: X_\tau \longmapsto X_\tau \cdot \Phi(X_\tau, \theta_{\text{enc}}) = X_\tau \cdot \hat{\Pi}_\tau \tag{8}$$

### 3.3 REORDERING TRAINING

Only joint-training the network $\hat{y} \circ \text{enc}$ as described above, will not teach CHAMELEON denoted by $\Phi$ how to reorder the features to a shared representation. That is why it is necessary to train $\Phi$ specifically with the objective of reordering features (reordering training). In order to do so, we optimize $\Phi$ to align novel tasks by training on a set of tasks for which the reordering matrix $\Pi_\tau$ exists such that it maps $\tau$ to the shared representation. In other words, we require a meta-dataset that contains not only a set of similar tasks $\tau \in \mathcal{T}$ with different schema, but also the position for each feature in the shared representation given by a permutation matrix. If $\Pi_\tau$ is known beforehand for each $\tau \in \mathcal{T}$, optimizing Chameleon becomes a simple supervised classification task based on predicting the new position of each feature in $\tau$. Thus, we can minimize the expected reordering loss over the meta-dataset:

$$\theta_{\text{enc}} = \underset{\theta_{\text{enc}}}{\arg\min} \, \mathbb{E}_{\tau \sim \mathcal{T}} L_\Phi\left(\Pi_\tau, \hat{\Pi}_\tau\right) \tag{9}$$

where $L_\Phi$ is the softmax cross-entropy loss, $\Pi_\tau$ is the ground-truth (one-hot encoding of the new position for each variable), and $\hat{\Pi}_\tau$ is the prediction. This training procedure can be seen in Algorithm (2). The trained CHAMELEON model can then be used to compute the $\Pi_\tau$ for any unseen task $\tau \in \mathcal{T}$.

---

**Algorithm 2** Reordering Training

---

**Input**: Meta-dataset $\mathcal{T} = \{(X_1, \Pi_1), ..., (X_{|\mathcal{T}|}, \Pi_{|\mathcal{T}|})\}$, latent dimension $K$, learning rate $\gamma$

1: Randomly initialize parameters $\theta_{\text{enc}}$ of the CHAMELEON model
2: **for** training iteration = 1, 2, ... **do**
3:     randomly sample $\tau \sim \mathcal{T}$
4:     $\theta_{\text{enc}} \longleftarrow \theta_{\text{enc}} - \gamma \nabla L_\Phi(\Pi_\tau, \Phi(X_\tau, \theta_{\text{enc}}))$
5: **end for**
6: **return** Trained parameters $\theta_{\text{enc}}$ of the CHAMELEON model

---

After this training procedure, we can use the learned weights as initialization for $\Phi$ before optimizing $\hat{y} \circ \text{enc}$ with REPTILE without further using $L_\Phi$. Experiments show that this procedure improves our results significantly compared to only optimizing the joint meta-loss.

Training the CHAMELEON component to reorder similar tasks to a shared representation not only requires a meta-dataset but one where the true reordering matrix $\Pi_\tau$ is provided for every task. In application, this means manually matching similar features of different training tasks so that novel tasks can be matched automatically. However, it is possible to sample a broad number of tasks from a

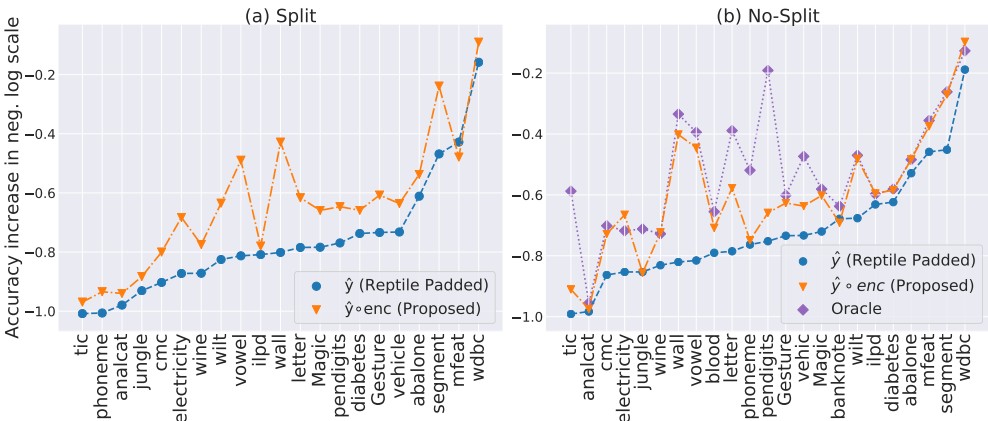

Figure 3: **Accuracy improvement for each method over Glorot initialization (Glorot & Bengio, 2010)**: The difference is plotted in negative log scale to account for the varying performance scales across the different datasets (higher points are better; A value of $-1$ is equivalent to the Glorot initialization). The graph (a) represents *Split* experiments while (b) depicts the *No-Split* experiments. Notice that the oracle has been omitted from the *Split* experiments since there is no true feature alignment for unseen features. The dataset axis is sorted by the performance of REPTILE on the base model to improve readability. All results are averaged over 5 runs.

single dataset by sampling smaller sub-tasks from it, selecting a random subset of features in arbitrary order for $N$ random instances. Thus, it is not necessary to manually match the features since all these sub-tasks share the same $\hat{\Pi}_\tau$ apart from the respective permutation of the rows as mentioned above.

## 4 EXPERIMENTAL RESULTS

**Baseline and Setup**   In order to evaluate the proposed method, we investigate the combined model $\hat{y} \circ \text{enc}$ with the initialization for enc obtained by pretraining CHAMELEON as defined in Equation 9 before using REPTILE to jointly optimize $\hat{y} \circ \text{enc}$. We compare the performance with an initialization obtained by running REPTILE on the base model $\hat{y}$ by training on tasks padded to a fixed size $K$ as $\hat{y}$ is not schema invariant. Both initializations are then compared to the performance of model $\hat{y}$ with random Glorot initialization (Glorot & Bengio, 2010) (referred to as *Random*). In all of our experiments, we measure the performance of a model and its initialization by evaluating the validation data of a task after performing three update steps on the respective training data. All experiments are conducted in two variants: In *Split* experiments, test tasks contain novel features in addition to features seen during meta-training. In contrast, test tasks in *No-Split* experiments only consist of features seen during meta-training. While the *Split* experiments evaluate the performance of the model when faced with novel features during meta-testing, the *No-Split* experiments can be used to compare against a perfect alignment by repeating the baseline experiment with tasks that are already aligned (referred to as *Oracle*). A detailed description of the utilized models is found in Appendix B.

**Meta-Datasets**   For our main experiments, we utilize a single dataset as meta-dataset by sampling the training and test tasks from it. This allows us to evaluate our method on different domains without matching related datasets since $\hat{\Pi}_\tau$ is naturally given for a subset of permuted features. Novel features can also be introduced during testing by splitting not only the instances but also the features of a dataset in train and test partition (*Split*). Training tasks are then sampled by selecting a random subset of the training features in arbitrary order for $N$ instances. Stratified sampling guarantees that test tasks contain both features from train and test while sampling the instances from the test set only. For all experiments, 75% of the instances are used for reordering training of CHAMELEON and joint-training of the full architecture, and 25% for sampling test tasks. For *Split* experiments, we further impose a train-test split on the features (20% of the features are restricted to the test split). Our work is built on top of REPTILE (Nichol et al., 2018b) but can be used in conjunction with any model-agnostic meta-learning method. We opted to use REPTILE since it does not require second-order derivatives, and the code is publicly available (Nichol et al., 2018a) while also being easy to adapt to our problem.

Figure 4: **Critical Difference Diagram** for *Split* (Left) and *No-Split* (Right) showing results of Wilcoxon signed-rank test with Holm's alpha correction and 5% significance level. Models are ranked by their performance and a thicker horizontal line indicates pairs that are not statistically different.

**Main Results**   We evaluate our approach using the OpenML-CC18 benchmark (Bischl et al., 2017) from which we selected 23 datasets for few-shot classification. The details of all datasets utilized in this work are summarized in Appendix B. The results in Figure 3 display the model performance after performing three update steps on a novel test task to illustrate the faster convergence. The graph shows a clear performance lift when using the proposed architecture after pretraining it to reorder tasks. This demonstrates to the best of our knowledge the first few-shot classification approach, which successfully learns across tasks with varying schemas (contribution 2). Furthermore, in the *No-Split* results one can see that the performance of the proposed method approaches the *Oracle* performance, which suggests an ideal feature alignment. When adding novel features during test time (*Split*) CHAMELEON is still able to outperform the other setups although with a lower margin.

**Ablations**   We visualize the result of pretraining CHAMELEON on the Wine dataset (from OpenML-CC18) in Figure 6 to show that the proposed model is capable of learning the correct alignment between tasks. One can see that the component manages to learn the true feature position in almost all cases. Moreover, this illustration does also show that CHAMELEON can be used to compute the similarity between different features by indicating which pairs are confused most often. For example, features two and four are showing a strong correlation, which is very plausible since they depict the *free sulfur dioxide* and *total sulfur dioxide level* of the wine. This demonstrates that our proposed architecture is able to learn an alignment between different feature spaces (contribution 1).

Furthermore, we repeat the experiments on the OpenML-CC18 benchmark in two ablation studies to measure the impact of joint-training and the proposed reordering training (Algorithm 2). First, we do not train CHAMELEON with Equation 9, but only jointly train $\hat{y} \circ \text{enc}$ with REPTILE to evaluate the influence of adding additional parameters to the network without pretraining it. Secondly, we use REPTILE only to update the initialization for the parameters of $\hat{y}$ while freezing the pretrained parameters of enc in order to assess the effect of joint-training both network components. These two variants are referred to as *Untrain* and *Frozen*. We compare these ablations to our approach by conducting a Wilcoxon signed-rank test (Wilcoxon, 1992) with Holm's alpha correction (Holm, 1979). The results are displayed in the form of a critical difference diagram (Demšar, 2006; Ismail Fawaz et al., 2019) presented in Figure 4. The diagram shows the ranked performance of each model and whether they are statistically different. The results confirm that our approach leads to statistically significant improvements over the random and REPTILE baselines when pretraining CHAMELEON. Similarly, our approach is also significantly better than jointly training the full architecture without pretraining CHAMELEON (UNTRAIN), confirming that the improvements do not stem from the increased model capacity. Finally, comparing the results to the FROZEN model shows improvements that are not significant, indicating that a near-optimal alignment was already found during pretraining. A detailed overview for all experimental results is given in Appendix C.

**Latent Embeddings Experiments**   Learning to align features is only feasible for unstructured data since this approach would not preserve any structure. However, it is a widespread practice among few-shot classification methods, and computer vision approaches in general, to use a pretrained model to embed image data into a latent space before applying further operations. We can use CHAMELEON to align the latent embeddings of image datasets that are generated with different networks. Thus, it is possible to use latent embeddings for meta-training while evaluating on novel tasks that are not yet embedded in case the embedding network is not available, or the complexity of different datasets requires models with different capacities to extract useful features. We conduct an additional experiment for which we combine two similar image datasets, namely *EMNIST-Digits* and *EMNIST-Letters* (Cohen et al., 2017). Similar to the work of Rusu et al. (2019), we train one neural network on each dataset in order to generate similar latent embeddings with different schema, namely 32 and 64 latent features. Afterward, we can sample training tasks from one embedding while

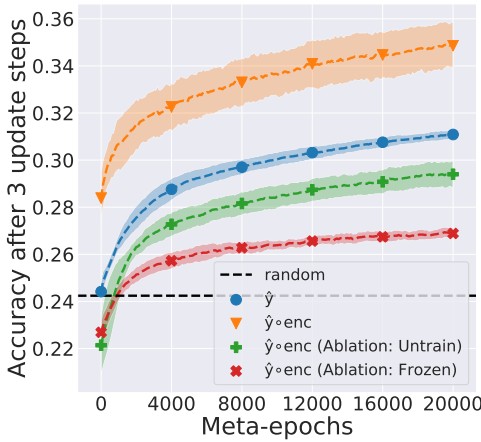

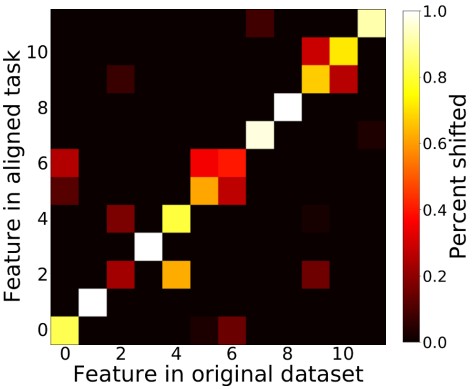

Figure 5: **Latent embedding results**. Meta test accuracy on the *EMNIST-Digits* data set while training on *EMNIST-Letters*. Each point represents the accuracy on the 1600 test tasks after performing three update steps on the respective training data. Results are averaged over 5 runs.

Figure 6: **Heat map of the feature shifts for the Wine data computed with CHAMELEON after reordering-training**: The x-axis represents the twelve features of the original dataset in the correct order and the y-axis shows which position these features are shifted to when presented in a permuted subset.

evaluating on tasks sampled from the other one. In the combined experiments, the full training is performed on the *EMNIST-Letters* dataset, while *EMNIST-Digits* is used for testing. Splitting the features is not necessary as the train, and test features are coming from different datasets. The results of this experiment are displayed in Figure 5. It shows the accuracy of *EMNIST-Digits* averaged across 5 runs with 1,600 generated tasks per run during the REPTILE training on *EMNIST-Letters* for the different model variants. Each test task is evaluated by performing 3 update steps on the training samples and measuring the accuracy of its validation data afterward. One can see that our proposed approach reports a significantly higher accuracy than the REPTILE baseline after performing three update steps on a task (contribution 4). Thus, showing that CHAMELEON is able to transfer knowledge from one dataset to another. Moreover, simply adding CHAMELEON without pretraining it to reorder tasks (*Untrain*) does not lead to any improvement. This might be sparked by using a CHAMELEON component that has a much lower number of parameters than the base network. Only by adding the reordering-training, the model manages to converge to a suitable initialization. In contrast to our experiments on the OpenML datasets, freezing the weights of CHAMELEON after pretraining also fails to give an improvement, suggesting that the pretraining did not manage to capture the ideal alignment, but enables learning it during joint-training. Our code is available at BLIND-REVIEW.

## 5 CONCLUSION

In this paper, we presented, to the best of our knowledge, the first approach to tackle few-shot classification for unstructured tasks with different schema. Our model component CHAMELEON is capable of embedding tasks to a common representation by computing a matrix that can reorder the features. For this, we propose a novel pretraining framework that is shown to learn useful permutations across tasks in a supervised fashion without requiring actual labels. In experiments on 23 datasets of the OpenML-CC18 benchmark, our method shows significant improvements even when presented with features not seen during training. Furthermore, by aligning different latent embeddings we demonstrate how a single meta-model can be used to learn across multiple image datasets each embedded with a distinct network.

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

# A    APPENDIX - INNER TRAINING

We visualize the inner training for one of the experiments in Figure 7. It shows two exemplary snapshots of the inner test loss when training on a sampled task with the current initialization $\theta^{\text{init}}$ before meta-learning and after 20,000 meta-epochs. It is compared to the test loss of the model when it is trained on the same task starting with the random initialization. For this experiment, models were trained until convergence. Note that both losses are not identical in meta-epoch 0 because the CHAMELEON component is already pretrained. The snapshots show the expected REPTILE behavior, namely a faster convergence when using the currently learned initialization compared to a random one.

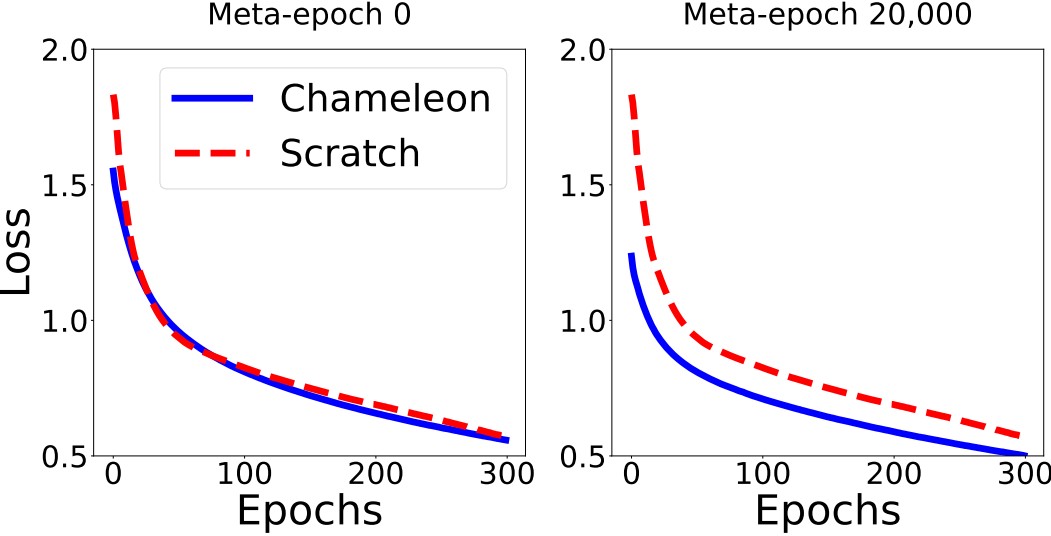

Figure 7: **Snapshots visualizing the inner training**. Validation cross-entropy loss for a task sampled from the *wall-robot-navigation* data set during inner training starting from the current initialization (blue) and random initialization (red).

# B    APPENDIX - EXPERIMENTAL DETAILS

The features of each dataset are normalized between 0 and 1. The *Split* experiments are limited to the 21 datasets which have more than four features in order to perform a feature split. We sample 10 training and 10 validation instances per label for a new task, and 16 tasks per meta-batch. The number of classes in a task is given by the number of classes of the respective dataset, as shown in Table 1. During the reordering-training phase and the inner updates of reptile, specified in line 6 of Algorithm (1), we use the ADAM optimizer (Kingma & Ba, 2014) with an initial learning rate of 0.0001 and 0.001 respectively. The meta-updates of REPTILE are carried out with a learning rate $\beta$ of 0.01. The reordering-training phase is run for 4000 epochs. All results reported in this work are averaged over 5 runs.

**OpenML-CC18**    All experiments on the OpenML-CC18 benchmark are conducted with the same model architecture. The base model $\hat{y}$ is a feed-forward neural network with two dense hidden layers that have 16 neurons each. CHAMELEON consists of two 1D-convolutions with 8 and 16 filters respectively and a final convolution that maps the task to the feature-length $K$, as shown in Figure 2. We selected dataasets that have up to 33 features and a minimum number of 90 instances per class. We limited the number of features and model capacity because this work seeks to establish a proof of concept for learning across data with different schemas. In contrast, very high-dimensional data would require tuning a more complex CHAMELEON architecture. The details for each dataset are summarized in Appendix 1. When sampling a task in *Split*, we sample between 40% and 60% of the respective training features. For test tasks in *Split* experiments 20% of the features are sampled from the set of test features to evaluate performance on similar tasks with partially novel features. For each

experimental run, the different variants are tested on the same data split, and we sample 1600 test tasks beforehand, while the training tasks are randomly sampled each epoch. All experiments are repeated five times with different instance and, in the case of *Split*, different feature splits, and the results are averaged.

**Latent Embeddings**   Both networks used for generating the latent embeddings consist of two convolutional and two dense hidden layers with 64 neurons each, but the number of neurons in the output layer is 32 for *EMNIST-Digits* and 64 for *EMNIST-Letters*. For these experiments, the CHAMELEON component still has two convolutional layers with 8 and 16 filters, while we use a larger base network with two feed-forward layers with 64 neurons each. All experimental results are averaged over five runs.

| Dataset | Instances | Features | Classes | Full Name |
|---------|-----------|----------|---------|-----------|
| phonem | 5404 | 5 | 2 | phoneme |
| cmc | 1473 | 24 | 3 | cmc |
| vowel | 990 | 27 | 11 | vowel |
| analcat | 797 | 21 | 6 | analcatdata-dmft |
| tic | 958 | 27 | 2 | tic-tac-toe |
| banknote | 1372 | 4 | 2 | banknote-authentication |
| wdbc | 569 | 30 | 2 | wdbc |
| diabetes | 768 | 8 | 2 | diabetes |
| segment | 2310 | 16 | 7 | segment |
| Magic | 19020 | 10 | 2 | MagicTelescope |
| blood | 748 | 4 | 2 | blood-transfusion-service-center |
| wall | 5456 | 24 | 4 | wall-robot-navigation |
| wilt | 4839 | 5 | 2 | wilt |
| pendigits | 10992 | 16 | 10 | pendigits |
| Gesture | 9873 | 32 | 5 | GesturePhaseSegmentationProcessed |
| abalone | 4177 | 10 | 3 | abalone |
| jungle | 44819 | 6 | 3 | jungle-chess-2pcs-raw-endgame-complete |
| letter | 20000 | 16 | 26 | letter |
| ilpd | 583 | 11 | 2 | ilpd |
| wine | 6497 | 11 | 5 | wine-quality |
| mfeat | 2000 | 6 | 10 | mfeat-morphological |
| electric | 45312 | 14 | 2 | electricity |
| vehicle | 846 | 18 | 4 | vehicle |
| **Embedded Datasets** | | | | |
| EDigits | 280,000 | 32* | 10 | EMNIST-Digits |
| ELetter | 145,600 | 64* | 28 | EMNIST-Letters |

Table 1: Information for the 23 OpenML-CC18 dataset used in this paper.* These datasets were embedded using our embedding neural network (see Apendix B).

## C   APPENDIX - TABLES WITH EXPERIMENTS RESULTS

The following tables show the detailed results of our experiments on the OpenML-CC18 datasets for *Split* and *NoSplit* settings. The tables contain the loss and accuracy for the the base model $\hat{y}$ trained from a random initialization and with REPTILE, and our proposed model $\hat{y} \circ \mathrm{enc}$ with the additional ablation studies *Untrain* and *Frozen*:

| | | | Loss | | | |
|---|---|---|---|---|---|---|
| Dataset | Random | $\hat{y}$ (Reptile Padded) | $\hat{y} \circ$ enc (Untrain) | $\hat{y} \circ$ enc (Proposed) | $\hat{y} \circ$ enc (Frozen) | $\hat{y}$ (Oracle) |
| segmen | 2.157 ± 0.003 | 1.409 ± 0.020 | 1.203 ± 0.056 | 0.928 ± 0.022 | **0.901 ± 0.030** | 0.940 ± 0.030 |
| jungle | 1.324 ± 0.004 | 1.079 ± 0.002 | 1.086 ± 0.002 | 1.081 ± 0.002 | **1.077 ± 0.002** | 1.023 ± 0.003 |
| wine | 1.851 ± 0.005 | 1.580 ± 0.003 | 1.567 ± 0.002 | **1.506 ± 0.006** | 1.513 ± 0.012 | 1.512 ± 0.009 |
| wilt | 0.848 ± 0.005 | 0.631 ± 0.002 | 0.653 ± 0.005 | 0.555 ± 0.008 | **0.549 ± 0.007** | 0.541 ± 0.004 |
| cmc | 1.327 ± 0.003 | 1.086 ± 0.003 | 1.057 ± 0.007 | **1.039 ± 0.002** | 1.042 ± 0.003 | 1.035 ± 0.007 |
| electr | 0.869 ± 0.004 | 0.686 ± 0.004 | 0.683 ± 0.002 | **0.639 ± 0.007** | 0.641 ± 0.007 | 0.655 ± 0.008 |
| letter | 3.426 ± 0.001 | 3.150 ± 0.020 | 3.033 ± 0.017 | 2.909 ± 0.024 | **2.689 ± 0.031** | 2.377 ± 0.031 |
| phonem | 0.858 ± 0.002 | **0.665 ± 0.005** | 0.684 ± 0.004 | 0.665 ± 0.005 | 0.668 ± 0.004 | 0.577 ± 0.005 |
| vehicl | 1.624 ± 0.004 | 1.310 ± 0.020 | 1.227 ± 0.008 | 1.214 ± 0.033 | **1.199 ± 0.037** | 1.063 ± 0.012 |
| mfeat | 2.535 ± 0.005 | 1.681 ± 0.031 | 1.486 ± 0.036 | **1.370 ± 0.027** | 1.405 ± 0.026 | 1.359 ± 0.049 |
| ilpd | 0.831 ± 0.006 | 0.626 ± 0.004 | 0.615 ± 0.003 | **0.603 ± 0.004** | 0.611 ± 0.003 | 0.605 ± 0.006 |
| Gestur | 1.809 ± 0.002 | 1.499 ± 0.006 | 1.437 ± 0.006 | **1.419 ± 0.002** | 1.421 ± 0.004 | 1.398 ± 0.006 |
| MagicT | 0.853 ± 0.002 | 0.649 ± 0.003 | 0.652 ± 0.008 | 0.604 ± 0.007 | **0.603 ± 0.004** | 0.590 ± 0.007 |
| tic | 0.871 ± 0.002 | 0.698 ± 0.001 | 0.694 ± 0.001 | **0.690 ± 0.000** | 0.690 ± 0.001 | 0.610 ± 0.004 |
| bankno | 0.840 ± 0.009 | 0.639 ± 0.004 | 0.654 ± 0.006 | **0.616 ± 0.001** | 0.621 ± 0.002 | 0.569 ± 0.003 |
| diabet | 0.851 ± 0.003 | 0.623 ± 0.002 | 0.638 ± 0.004 | 0.605 ± 0.002 | **0.598 ± 0.003** | 0.600 ± 0.004 |
| wdbc | 0.823 ± 0.010 | 0.311 ± 0.026 | 0.221 ± 0.014 | **0.158 ± 0.007** | 0.194 ± 0.014 | 0.197 ± 0.013 |
| blood | 0.845 ± 0.004 | 0.681 ± 0.003 | 0.688 ± 0.002 | 0.660 ± 0.003 | **0.659 ± 0.002** | 0.647 ± 0.001 |
| vowel | 2.641 ± 0.003 | 2.315 ± 0.015 | 1.912 ± 0.016 | **1.821 ± 0.023** | 1.843 ± 0.021 | 1.671 ± 0.029 |
| pendig | 2.545 ± 0.004 | 2.189 ± 0.006 | 2.169 ± 0.010 | 2.107 ± 0.020 | **2.099 ± 0.021** | 1.068 ± 0.034 |
| wall | 1.638 ± 0.003 | 1.360 ± 0.011 | 1.083 ± 0.014 | **0.972 ± 0.014** | 0.986 ± 0.007 | 0.868 ± 0.016 |
| abalon | 1.311 ± 0.003 | 0.871 ± 0.005 | 0.894 ± 0.009 | **0.828 ± 0.004** | 0.834 ± 0.005 | 0.823 ± 0.007 |
| analca | 2.062 ± 0.002 | 1.801 ± 0.000 | **1.794 ± 0.001** | 1.806 ± 0.002 | 1.806 ± 0.001 | 1.827 ± 0.004 |

| | | | Accuracy | | | |
|---|---|---|---|---|---|---|
| Dataset | Random | $\hat{y}$ (Reptile Padded) | $\hat{y} \circ$ enc (Untrain) | $\hat{y} \circ$ enc (Proposed) | $\hat{y} \circ$ enc (Frozen) | $\hat{y}$ (Oracle) |
| segmen | 0.147 ± 0.001 | 0.419 ± 0.005 | 0.496 ± 0.015 | 0.595 ± 0.012 | **0.619 ± 0.015** | 0.605 ± 0.009 |
| jungle | 0.335 ± 0.002 | 0.395 ± 0.003 | 0.382 ± 0.003 | 0.393 ± 0.004 | **0.396 ± 0.002** | 0.460 ± 0.003 |
| wine | 0.201 ± 0.002 | 0.264 ± 0.003 | 0.273 ± 0.003 | **0.314 ± 0.005** | 0.308 ± 0.007 | 0.312 ± 0.009 |
| wilt | 0.504 ± 0.002 | 0.628 ± 0.002 | 0.601 ± 0.009 | 0.718 ± 0.005 | **0.720 ± 0.005** | 0.724 ± 0.003 |
| cmc | 0.331 ± 0.002 | 0.386 ± 0.004 | 0.422 ± 0.008 | **0.448 ± 0.003** | 0.446 ± 0.002 | 0.461 ± 0.008 |
| electr | 0.499 ± 0.002 | 0.548 ± 0.010 | 0.559 ± 0.004 | **0.626 ± 0.011** | 0.625 ± 0.007 | 0.603 ± 0.011 |
| letter | 0.039 ± 0.000 | 0.078 ± 0.005 | 0.112 ± 0.005 | 0.153 ± 0.006 | **0.204 ± 0.006** | 0.282 ± 0.012 |
| phonem | 0.504 ± 0.003 | 0.594 ± 0.012 | 0.561 ± 0.005 | **0.600 ± 0.008** | 0.597 ± 0.004 | 0.702 ± 0.001 |
| vehicl | 0.255 ± 0.001 | 0.366 ± 0.010 | 0.413 ± 0.010 | 0.418 ± 0.022 | **0.434 ± 0.027** | 0.523 ± 0.009 |
| mfeat | 0.104 ± 0.002 | 0.354 ± 0.006 | 0.398 ± 0.008 | 0.428 ± 0.012 | **0.431 ± 0.009** | 0.447 ± 0.012 |
| ilpd | 0.506 ± 0.003 | 0.654 ± 0.005 | 0.659 ± 0.004 | **0.670 ± 0.005** | 0.662 ± 0.006 | 0.669 ± 0.006 |
| Gestur | 0.202 ± 0.002 | 0.310 ± 0.002 | 0.350 ± 0.006 | **0.368 ± 0.003** | 0.364 ± 0.004 | 0.383 ± 0.002 |
| MagicT | 0.503 ± 0.002 | 0.611 ± 0.002 | 0.601 ± 0.012 | **0.662 ± 0.007** | 0.661 ± 0.005 | 0.672 ± 0.004 |
| tic | 0.502 ± 0.002 | 0.504 ± 0.003 | 0.510 ± 0.003 | 0.533 ± 0.001 | **0.534 ± 0.005** | 0.666 ± 0.005 |
| bankno | 0.506 ± 0.005 | **0.634 ± 0.005** | 0.622 ± 0.003 | 0.629 ± 0.004 | 0.626 ± 0.004 | 0.652 ± 0.003 |
| diabet | 0.505 ± 0.004 | 0.656 ± 0.004 | 0.639 ± 0.007 | **0.674 ± 0.002** | 0.673 ± 0.003 | 0.675 ± 0.006 |
| wdbc | 0.521 ± 0.007 | 0.882 ± 0.008 | 0.906 ± 0.008 | **0.937 ± 0.003** | 0.918 ± 0.007 | 0.919 ± 0.006 |
| blood | 0.502 ± 0.001 | 0.579 ± 0.012 | 0.558 ± 0.010 | 0.613 ± 0.007 | **0.615 ± 0.003** | 0.636 ± 0.004 |
| vowel | 0.092 ± 0.001 | 0.143 ± 0.007 | 0.303 ± 0.007 | **0.346 ± 0.010** | 0.336 ± 0.008 | 0.391 ± 0.013 |
| pendig | 0.102 ± 0.001 | 0.180 ± 0.003 | 0.193 ± 0.004 | 0.222 ± 0.011 | **0.227 ± 0.009** | 0.646 ± 0.010 |
| wall | 0.254 ± 0.001 | 0.324 ± 0.012 | 0.494 ± 0.007 | **0.576 ± 0.009** | 0.562 ± 0.007 | 0.631 ± 0.010 |
| abalon | 0.339 ± 0.003 | 0.566 ± 0.002 | 0.554 ± 0.007 | **0.594 ± 0.004** | 0.587 ± 0.004 | 0.593 ± 0.005 |
| analca | 0.166 ± 0.001 | 0.170 ± 0.000 | 0.170 ± 0.002 | **0.172 ± 0.002** | 0.171 ± 0.002 | 0.179 ± 0.002 |

Table 2: Loss and accuracy scores of each model variant for the *No-Split* experiments. The values depict the mean and standard deviation across 5 runs for each dataset with 1600 sampled test tasks per run. Best results are boldfaced (excluding ORACLE).

| | | | Loss | | |
|---|---|---|---|---|---|
| Dataset | Random | $\hat{y}$ (Reptile Padded) | $\hat{y} \circ$ enc (Untrain) | $\hat{y} \circ$ enc (Proposed) | $\hat{y} \circ$ enc (Frozen) |
| vowel | $2.640 \pm 0.001$ | $2.313 \pm 0.007$ | $1.969 \pm 0.016$ | $1.913 \pm 0.013$ | $\mathbf{1.911 \pm 0.016}$ |
| wdbc | $0.826 \pm 0.014$ | $0.264 \pm 0.031$ | $0.167 \pm 0.010$ | $\mathbf{0.162 \pm 0.002}$ | $0.170 \pm 0.006$ |
| jungle | $1.332 \pm 0.004$ | $1.142 \pm 0.014$ | $\mathbf{1.089 \pm 0.002}$ | $1.099 \pm 0.005$ | $1.093 \pm 0.004$ |
| phonem | $0.856 \pm 0.003$ | $0.769 \pm 0.034$ | $0.719 \pm 0.005$ | $0.720 \pm 0.010$ | $\mathbf{0.716 \pm 0.009}$ |
| wine | $1.855 \pm 0.002$ | $1.596 \pm 0.004$ | $1.582 \pm 0.005$ | $\mathbf{1.546 \pm 0.018}$ | $1.547 \pm 0.013$ |
| analca | $2.061 \pm 0.001$ | $1.802 \pm 0.002$ | $\mathbf{1.794 \pm 0.001}$ | $1.796 \pm 0.001$ | $1.798 \pm 0.002$ |
| MagicT | $0.851 \pm 0.004$ | $0.673 \pm 0.009$ | $0.662 \pm 0.002$ | $\mathbf{0.629 \pm 0.007}$ | $0.630 \pm 0.004$ |
| diabet | $0.850 \pm 0.004$ | $0.675 \pm 0.009$ | $0.677 \pm 0.008$ | $\mathbf{0.646 \pm 0.012}$ | $0.655 \pm 0.014$ |
| letter | $3.426 \pm 0.001$ | $3.160 \pm 0.017$ | $3.058 \pm 0.009$ | $2.980 \pm 0.031$ | $\mathbf{2.782 \pm 0.032}$ |
| ilpd | $0.840 \pm 0.002$ | $0.692 \pm 0.005$ | $\mathbf{0.689 \pm 0.008}$ | $0.694 \pm 0.004$ | $0.694 \pm 0.004$ |
| Gestur | $1.813 \pm 0.003$ | $1.514 \pm 0.006$ | $1.429 \pm 0.004$ | $\mathbf{1.413 \pm 0.005}$ | $1.416 \pm 0.003$ |
| mfeat | $2.531 \pm 0.005$ | $1.591 \pm 0.067$ | $\mathbf{1.417 \pm 0.010}$ | $1.627 \pm 0.053$ | $1.620 \pm 0.059$ |
| wilt | $0.844 \pm 0.003$ | $0.721 \pm 0.034$ | $0.652 \pm 0.007$ | $\mathbf{0.633 \pm 0.026}$ | $0.671 \pm 0.019$ |
| wall | $1.640 \pm 0.004$ | $1.356 \pm 0.002$ | $1.081 \pm 0.009$ | $\mathbf{0.993 \pm 0.010}$ | $1.003 \pm 0.009$ |
| segmen | $2.166 \pm 0.002$ | $1.388 \pm 0.061$ | $1.147 \pm 0.021$ | $\mathbf{0.799 \pm 0.024}$ | $0.840 \pm 0.020$ |
| cmc | $1.327 \pm 0.001$ | $1.098 \pm 0.003$ | $1.086 \pm 0.003$ | $\mathbf{1.076 \pm 0.011}$ | $1.082 \pm 0.004$ |
| pendig | $2.548 \pm 0.003$ | $2.210 \pm 0.016$ | $2.195 \pm 0.015$ | $2.123 \pm 0.009$ | $\mathbf{2.038 \pm 0.196}$ |
| electr | $0.865 \pm 0.003$ | $0.691 \pm 0.005$ | $0.686 \pm 0.001$ | $\mathbf{0.642 \pm 0.005}$ | $0.646 \pm 0.007$ |
| vehicl | $1.624 \pm 0.005$ | $1.289 \pm 0.008$ | $1.221 \pm 0.004$ | $\mathbf{1.193 \pm 0.018}$ | $1.225 \pm 0.004$ |
| abalon | $1.313 \pm 0.004$ | $0.971 \pm 0.025$ | $0.929 \pm 0.004$ | $\mathbf{0.894 \pm 0.014}$ | $0.910 \pm 0.003$ |
| tic | $0.870 \pm 0.003$ | $0.703 \pm 0.003$ | $0.695 \pm 0.001$ | $0.696 \pm 0.002$ | $\mathbf{0.694 \pm 0.001}$ |

| | | | Accuracy | | |
|---|---|---|---|---|---|
| Dataset | Random | $\hat{y}$ (Reptile Padded) | $\hat{y} \circ$ enc (Untrain) | $\hat{y} \circ$ enc (Proposed) | $\hat{y} \circ$ enc (Frozen) |
| vowel | $0.092 \pm 0.001$ | $0.144 \pm 0.003$ | $0.288 \pm 0.007$ | $\mathbf{0.311 \pm 0.009}$ | $0.311 \pm 0.007$ |
| wdbc | $0.522 \pm 0.009$ | $0.901 \pm 0.011$ | $0.937 \pm 0.004$ | $\mathbf{0.942 \pm 0.003}$ | $0.935 \pm 0.004$ |
| jungle | $0.333 \pm 0.001$ | $0.359 \pm 0.011$ | $0.385 \pm 0.004$ | $0.378 \pm 0.009$ | $\mathbf{0.388 \pm 0.005}$ |
| phonem | $0.503 \pm 0.002$ | $0.504 \pm 0.021$ | $0.502 \pm 0.017$ | $0.529 \pm 0.004$ | $\mathbf{0.533 \pm 0.024}$ |
| wine | $0.201 \pm 0.002$ | $0.248 \pm 0.004$ | $0.265 \pm 0.007$ | $\mathbf{0.289 \pm 0.010}$ | $0.285 \pm 0.010$ |
| analca | $0.167 \pm 0.001$ | $0.172 \pm 0.003$ | $0.173 \pm 0.002$ | $\mathbf{0.185 \pm 0.002}$ | $0.182 \pm 0.002$ |
| MagicT | $0.502 \pm 0.002$ | $0.582 \pm 0.010$ | $0.586 \pm 0.003$ | $0.634 \pm 0.010$ | $\mathbf{0.634 \pm 0.004}$ |
| diabet | $0.501 \pm 0.002$ | $0.601 \pm 0.012$ | $0.605 \pm 0.008$ | $\mathbf{0.635 \pm 0.012}$ | $0.635 \pm 0.017$ |
| letter | $0.039 \pm 0.000$ | $0.080 \pm 0.005$ | $0.108 \pm 0.003$ | $0.137 \pm 0.007$ | $\mathbf{0.181 \pm 0.006}$ |
| ilpd | $0.501 \pm 0.002$ | $0.571 \pm 0.004$ | $0.579 \pm 0.004$ | $\mathbf{0.583 \pm 0.006}$ | $0.580 \pm 0.005$ |
| Gestur | $0.200 \pm 0.001$ | $0.306 \pm 0.003$ | $0.361 \pm 0.003$ | $\mathbf{0.375 \pm 0.004}$ | $0.372 \pm 0.004$ |
| mfeat | $0.103 \pm 0.001$ | $0.377 \pm 0.024$ | $\mathbf{0.425 \pm 0.023}$ | $0.336 \pm 0.027$ | $0.335 \pm 0.015$ |
| wilt | $0.504 \pm 0.004$ | $0.563 \pm 0.043$ | $0.598 \pm 0.011$ | $\mathbf{0.643 \pm 0.034}$ | $0.589 \pm 0.030$ |
| wall | $0.252 \pm 0.002$ | $0.330 \pm 0.004$ | $0.487 \pm 0.005$ | $\mathbf{0.553 \pm 0.004}$ | $0.543 \pm 0.005$ |
| segmen | $0.148 \pm 0.002$ | $0.414 \pm 0.022$ | $0.501 \pm 0.013$ | $\mathbf{0.638 \pm 0.009}$ | $0.617 \pm 0.013$ |
| cmc | $0.333 \pm 0.001$ | $0.371 \pm 0.004$ | $0.394 \pm 0.006$ | $\mathbf{0.415 \pm 0.013}$ | $0.408 \pm 0.006$ |
| pendig | $0.102 \pm 0.001$ | $0.173 \pm 0.005$ | $0.181 \pm 0.008$ | $0.229 \pm 0.002$ | $\mathbf{0.257 \pm 0.079}$ |
| electr | $0.500 \pm 0.002$ | $0.545 \pm 0.006$ | $0.551 \pm 0.002$ | $\mathbf{0.622 \pm 0.007}$ | $0.616 \pm 0.009$ |
| vehicl | $0.254 \pm 0.002$ | $0.369 \pm 0.008$ | $0.397 \pm 0.007$ | $\mathbf{0.420 \pm 0.015}$ | $0.397 \pm 0.010$ |
| abalon | $0.338 \pm 0.002$ | $0.513 \pm 0.016$ | $0.532 \pm 0.003$ | $\mathbf{0.556 \pm 0.010}$ | $0.543 \pm 0.003$ |
| tic | $0.501 \pm 0.002$ | $0.501 \pm 0.001$ | $0.506 \pm 0.003$ | $0.514 \pm 0.002$ | $\mathbf{0.515 \pm 0.003}$ |

Table 3: Loss and accuracy scores of each model variant for the *Split* experiments. The values depict the mean and standard deviation across 5 runs for each dataset with 1600 sampled test tasks per run. Best results are boldfaced.

## D PROBLEM SETTING: GENERAL MULTI-TASK LEARNING.

We describe a classification dataset with vector-shaped predictors (i.e., no images, time series etc.) by a pair $(X, Y) \in \mathbb{R}^{N \times F} \times \{0, ..., C\}^N$, with predictors $X$ and targets $Y$, where $N$ denotes the number of instances, $F$ the number of predictors and $C$ the number of classes. Let $\mathcal{D}_F := \bigcup_{N \in \mathbb{N}} \mathbb{R}^{N \times F} \times \{0, ..., C\}^N$ be the space of all such datasets with $F$ predictors and $\mathcal{D} := \bigcup_{F \in \mathbb{N}} \mathcal{D}_F$ be the space of any such dataset. Let us also denote the space of all predictor matrices with $F$ predictors by $\mathcal{X}_F := \bigcup_{N \in \mathbb{N}} \mathbb{R}^{N \times F}$ and all predictor matrices by $\mathcal{X} := \bigcup_{F \in \mathbb{N}} \mathcal{X}_F$. Then a dataset $\tau = (X, Y) \in \mathcal{D}$ equipped with a predefined training/test split, i.e. the quadruplet $\tau = (X_\tau^{\text{train}}, Y_\tau^{\text{train}}, X_\tau^{\text{test}}, Y_\tau^{\text{test}})$ is called a *task*. A collection of such tasks $\mathcal{T} \subset \mathcal{D}$ is called a *meta-dataset*. Similar to splitting a single data set into a training and test part, one can split a meta-dataset $\mathcal{T} = \mathcal{T}^{\text{train}} \mathbin{\dot{\cup}} \mathcal{T}^{\text{test}}$.

Consider a meta-dataset of correlated tasks $\mathcal{T} \subset \mathcal{D}$, such that the predictor variables $\{p_{\tau_1}, p_{\tau_2}, \ldots, p_{\tau_F}\}$ of any individual task $\tau$ are contained in a common set of predictor variables $\{p_1, p_2, \ldots, p_K\}$. As elucidated in the previous section, our goal is to construct an encoder that learns to match these predictors and map the features of any task $\tau \in \mathcal{T}$ into a shared latent space $\mathbb{R}^K$.

$$\text{enc}: \mathcal{X} \longrightarrow \mathcal{X}_K, \ X \in \mathbb{R}^{N \times F} \longmapsto \widetilde{X} \in \mathbb{R}^{N \times K} \tag{10}$$

This encoder can be combined with a parametric model of fixed input size $\hat{y}: \mathbb{R}^K \to \{0, 1\}$ (e.g. neural network or SVM) such that for the joint model $\hat{y} \circ \text{enc}$ an initialization $\theta^{\text{init}}$ can be learned via MAML or REPTILE across all tasks, even when those may not have the same predictor vector. Just as with MAML, this initialization facilitates rapid convergence of the combined model $\hat{y} \circ \text{enc}$ on any new, previously unseen task $T \in \mathcal{T}^{\text{test}}$. More explicitly, the ultimate goal is to minimize the meta test loss

$$\mathcal{L}(\theta^{\text{init}}) := \mathbb{E}_{T_\tau \sim \mathcal{T}^{\text{test}}} L_\tau \left( Y_\tau^{\text{test}}, \ \hat{y} \circ \text{enc} \left( X_\tau^{\text{test}}; \theta_\tau^{(u)} \right) \right) \tag{11}$$

here $L_\tau$ is the task specific loss (e.g. miss-classification rate) of the model on the test data of $T_\tau$, using the updated parameters $\theta_\tau^{(u)}$. The latter are the updated parameters of the joint model $\hat{y} \circ \text{enc}$ which are obtained by minimizing $L_\tau$ on the training data $(X^{\text{train}}, Y^{\text{train}})$ of $T_\tau$ via some learning iterative learning algorithm $\mathcal{A}$ (e.g. Gradient Descent) for $u$ iterations.

$$\theta_\tau^{(u)} = \mathcal{A}^{(u)} \left( X_\tau^{\text{train}}, Y_\tau^{\text{train}}, L_\tau, \hat{y} \circ \text{enc}; \theta^{\text{init}} \right) \tag{12}$$

MAML and REPTILE are solving sub-problems when the number $F$ of features is fixed and the predictors of all tasks are the same and aligned, i.e., the same predictor always occurs at the same position within the predictor vector, thus the identity can be used as predictor encoder. This problem alternatively can be described as a supervised learning problem with a multivariate or structured target.

