# OpenReview forum: "Chameleon: Learning Model Initializations Across Tasks With Different Schemas"
_ICLR.cc/2021/Conference — Reject_

### Official Review · AnonReviewer1 · 2020-10-28

**Rating:** 6
**Confidence:** 3

**Review:**

Previous meta-learning approaches typically focus on tasks that share the same input types, e.g. images.
This paper addresses the problem of meta-learning weight initialization across tasks with different types of input features.
It proposes Chameleon model that learns to align input features from different tasks by learning a permutation matrix for each task, and shows that Chameleon can successfully learn good initialization.


Strength:
- It identifies and tackles a new important problem in meta-learning: meta-learning on tasks with different input features.
- The proposed approach is simple but shows improvements over the baseline method.

Weakeness:
- Supervised training for the permutation matrix is necessary for the model to perform well.
- Experimental results section can be more detailed. Given that Algorithm 2 is the major part of the method, how is the reordering training procedure constructed? How is the target permutation matrix determined? are there / what are the shared features between different tasks?
- Would be great if experiments are done on one or two more datasets to strengthen the result.

Additional Comments:
- How many features are used? How would the performance change if there are more/fewer features?
- typo: Equation (9) is mentioned several times

I believe this paper proposed a new interesting problem in meta-learning and provided a simple effective model to address the problem.

---

> ### Author Response · Authors · 2020-11-14
> **Clarifications**
>
> Thank you for your insightful review. You are correct regarding the fact that Chameleon requires a ground-truth permutation matrix during reordering training. We show that it is possible to use a single dataset as meta-dataset to sample training tasks which means that the ground-truth permutation matrix for each sampled task is naturally given. A training task is sampled by selecting a subset of the features and instances from the meta-dataset so that the true permutation matrix is given by the original feature positions. Reordering training is then simple supervised learning on the given ground-truth permutation matrix as defined in Algorithm 2 by optimizing.
>
> For our OpenML datasets, we randomly select a subset of features that are excluded from meta-training and appear in test tasks. In the EMNIST experiment, the training and test tasks are sampled from two non-overlapping datasets. However, we can assume that due to the similarity of the datasets and embedding networks, there are latent features in both datasets which represent similar information and can thus be aligned which is why we can observe the significant lift in Figure 5.
>
> Regarding experimenting with bigger data-sets, we have conducted the EMNIST experiment for that purpose where two distinct datasets EMNIST Digits and EMNIST Letters with similar characteristics are used for meta-training and meta-testing. In the future, we want to expand upon this and train on larger meta-datasets with more diverse test tasks. However, this work is the first approach that learns an initialization across tasks with different schemas for unstructured data and thus operates as the first proof of concept.
>
> Thank you for pointing out the typo of Equation (9) which should be Equation (8) (It is now still Equation (9) because we've added an additional equation in section 2).

---

### Official Review · AnonReviewer4 · 2020-10-28
**Official Blind Review**

**Rating:** 6
**Confidence:** 3

**Review:**

Chameleon: Learning Model Initializations Across Tasks With Different Schemas


The paper provides an interesting direction in the few-shot classification field. In particular, it proposes a model that learns to align different predictor schemas to a common representation. The paper also demonstrates how current meta-learning approaches can successfully learn a model initialisation across tasks with different schemas as long as they share some variables with respect to their type or semantics.

The paper takes on an interesting facet of few-shot classification: An encoder model that aligns to different predictor schemas to a common representation. It tackles the problem by using 1D convolution (three of them) to transform the input features to the K-features target space and learning the alignment from the data itself. Comprehensive experiments have been done with quantitative results and analysis, to show the effectiveness of the proposed approach and the results are convincing and the code is provided to determine the reproducibility of the results.

Overall performance is quite good however, it would be a good study to have an analysis of the different datasets as to how balanced/unbalanced they are, how it affects the performance, the nature of the features etc. Also, I would like the author to discuss how suitable/adaptable this approach will be for multi-label tasks and what kind of modifications (if any) are to be made.

The idea of encoding different predictor schemas to a common representation is quite interesting and comprehensive experiments and supporting ablation study has been made.

---

> ### Author Response · Authors · 2020-11-14
> **Clarifications and considerations**
>
> Thank you for your great review. Further analysis of the side effects of characteristics of datasets such as balance and size is indeed an important aspect to be explored and it is one of our future works. Regarding the suitability for multi-label problems, our approach is generally model-agnostic and thus could be adapted to multi-label tasks with no further adjustments, as the only necessary changes required would occur in the utilized base model $\hat{y}$ and task-dependent loss function $L_\tau$.
>
> At the same time, Chameleon can be easily combined with other meta-learning methods that are specifically designed for multi-label classification or handle a varying number of classes such as LEO [1] or HIDRA [2]. In other words, Chameleon can be used for any popular meta-learning objective.
>
> [1] Andrei A. Rusu, Dushyant Rao, Jakub Sygnowski, Oriol Vinyals, Razvan Pascanu, Simon Osindero, and Raia Hadsell. Meta-learning with latent embedding optimization. In International Conference on Learning Representations, 2019
>
> [2] Rafael Rego Drumond, Lukas Brinkmeyer, Josif Grabocka, and Lars Schmidt-Thieme. HIDRA: Head initialization across dynamic targets for robust architectures. In Proceedings of the 2020 SIAM International Conference on Data Mining, pp. 397–405. SIAM, 2020.

---

### Official Review · AnonReviewer3 · 2020-10-29
**The technical contribution is limited and impractical.**

**Rating:** 4
**Confidence:** 4

**Review:**

- Summary and contributions
    - In this work, the authors tried to solve the problem of ``heterogeneous'' meta-learning where each task resides in a different feature space from the other tasks. They introduced a feature transformation or re-ordering matrix to align the features. While I agree with the authors that this problem is of significance in the meta-learning community, the solution in this work, depending on the ground-truth of re-ordering matrix, is trivial and impractical.

- Strengths:
    - The problem investigated in this paper, i.e., meta-learning tasks in heterogeneous feature spaces, is important to the field of meta-learning.
    - The paper is well written and easy to follow.

- Weaknesses:
    - The primary concern about this paper is its technical contribution, being limited and impractical. To align tasks in incommensurable feature spaces, projecting them into a common feature space has been a common practice. Please kindly see related works on heterogeneous transfer learning. The major challenge lies in the supervision needed to train the alignment matrix or function. The ground-truth feature alignment matrix is almost impractical to collect, if the dimension of features is super large and we have no knowledge of the semantic correspondence between two features from two tasks.
    - The empirical results are also not convincing.
         - Why is only Glorot initialization compared in Figure 3? What has been widely adopted is some better initialization strategies, including (He initialization).
        - From both Figure 3 and Figure 4, and also the results in Appendix C, I see little improvement of the proposed over Frozen. This means that most benefits of the feature alignment come from the supervised training part where a ground-truth alignment matrix is required to train $\Phi$, while the matrix is even infeasible to have in practical settings.
        - In Line 6 of the section "Ablations", the authors mentioned that features 2 and 3 are showing a strong correlation, but I cannot see why in Figure 6. Maybe it is features 2 and 4?

- Minor:
    Line 2 in Section 4: Equation (9) does not exist…

---

> ### Author Response · Authors · 2020-11-14
> **Discussion of contribution and clarifications**
>
> Thank you for your great feedback. We appreciate the input regarding heterogeneous transfer learning. The updated related work section includes a paragraph about recent and popular approaches that are related to our work. However, none of these approaches are capable of training a single encoder that operates across a meta-dataset of tasks with varying schemas for unstructured data. Most approaches tend to operate on structured data which can be embedded by existing approaches such as convolutional neural networks for images or transformer-based language models for text, while also utilizing characteristics like co-occurrence data, meta-features, or separate networks. Furthermore, these approaches are usually not designed for handling tasks with few samples.
>
> Regarding the empirical results, we have used Glorot initialization as it is often used as a default initialization in research. The main focus of our experiments is on the comparison of applying meta-learning few-shot approaches with or without learning a common alignment, while the randomly initialized model serves as a rudimentary baseline and sanity check. We can assume that no classical random initialization would show a better performance than the learned initialization on imputed tasks via Reptile which can be observed in our experiments. Internally, we did evaluate He initialization on a smaller subset of our experiments and observed a small lift of around 1-2 percent in accuracy compared to Glorot on average. However, this lift translated to all our results since we select the same initialization not only for the random baseline but also as a starting point for meta-learning. Thus, selecting Glorot or He has no further impact on the results of our work.
>
> Furthermore, it is true that the improvement over Frozen is not significant for the OpenML experiments. This indicates that a near-optimal alignment was already found during pretaining. However, we show that by utilizing a single dataset as a meta-dataset, one does not need to manually construct a ground-truth alignment matrix. We demonstrate in the EMNIST experiments, how this can be used to transfer knowledge between two similar datasets. In practice, the training dataset can be chosen as a large-scale dataset while the number of datasets in inference is not limited. Moreover, once can see that in the more complex EMNIST experiments the best alignment is only found by training the model on the classification objective, leading to significant improvements over Frozen.
>
> The strong correlation between feature 2 and 3 in Figure 6 was indeed a typo we fixed.
>
> Generally, we want to express that while this is a relatively simplistic approach, this is the first approach to our knowledge to tackle few-shot classification for unstructured data with different schemas and thus, we purposely wanted to introduce a simple model that can tackle the problem and show it is possible to learn across these tasks as the first proof of concept which also motivates future research.

---

### Official Review · AnonReviewer2 · 2020-10-30
**The idea of learning to re-align input spaces in a common feature space has merit, but the experimental protocol is unusual and results not convincing**

**Rating:** 3
**Confidence:** 5

**Review:**

Summary
--------------
The paper proposes a trainable way to re-order or recover the ordering of features from sets of examples, and use it as a way to build a common feature space (or embedding) for a neural net, the (initial) parameters of which can be trained by Reptile.
Experiments show that such initial parameters enable faster training (inside of an episode) than untrained weights.

Pros
------
- The paper shows it is possible to recover information about the identity of coordinates in the input space, through a learned transformation, on several unstructured datasets. The similarity between such representations of individual coordinates can help identify similar features, either in a given dataset or across datasets.


Cons
--------
The paper is overall really hard to follow, statements are often confusing or misleading. For instance:
- The introduction suggests a multi-modal learning paradigm, where different tasks could have access to data in different input spaces, some of them common. However, the paper then seems to consider individual coordinates in the input space only, and focuses on mapping shuffled subsets of these coordinates back to their initial position.
- There is confusion about the "tasks", which sometimes correspond to one of the OpenML datasets, and sometimes to individual few-shot episodes from one of these datasets.
- Concepts like "schema" and "predictors" are never properly introduced or defined.
- The description of the "chameleon" (alignment) component mentions "order-invariant" and "permutation invariant" several times, but it is quite unclear whether it refers to the the order of the examples within the data set (or episode) or the order in which the features are represented.

The paper uses few-shot learning vocabulary and techniques, including Reptile, but the methodology seems completely different from the few-shot learning literature. In particular:
- There does not appear to be a split between meta-training and meta-test classes within a dataset, or meta-training datasets and meta-testing ones, except for the EMNIST experiment. Even then, the pre-training of the "chameleon" alignment module seems to involve using examples of the meta-test classes.
- The reported evaluation metric is really unusual: they report the improvement (and sometimes accuracy) after 3 steps of gradient descent from within an episode, which is somewhat related to the quality of the meta-learned weights, but no other metric that would be comparable to existing literature, which makes it especially hard to assess the results.

The principle of the alignment module seems similar to (soft) attention mechanisms, in that there is a softmax trained to highlight which parts of an input vector should be emphasized (or selected) at a given point in the processing (here, in the aligned feature space). However, the literature on attention is not reviewed.

Many design choices are not addressed clearly, neither in how they were made, or the impact of these choices, especially regarding the architecture of the alignment module:
- It is a linear transformation (before the softmax), though parameterized by 3 matrices. An alternative would have been a 3-layer neural net, similar to attention networks.
- The parameterization of the first matrix makes the number of parameters depend on N, the number of examples in a given task. This could be quite limiting to be restrained to tasks of exactly N examples, especially if both the support (mini-train) and query (mini-test or valid) parts of an episode need to have exactly N examples.
- There is also no discussion of the  value or impact of or K, the size of the chosen embedding space).

Recommendation
--------------------------
I recommend to reject this submission.

Arguments
------------------
The main idea in the paper, learning alignments of various input spaces into a common embedding space through an attention mechanism, has merit and may  work reasonably.
However, both the algorithm and the experimental set up are described in a quite confused way, and not well justified or grounded. The reported results are not comparable with few-shot learning literature, nor multi-modal training or feature imputation, and do not make a convincing case.

Questions
---------------
As I understand it, the "Chameleon" architecture itself simply consists in 3 matrix multiplications (Nx8, 8x16, 16xK), which would be equivalent to the length-1 1D convolutions, is that correct? It may be more straightforward to explain that way, as $enc(X) = X M_1 M_2 M_3 X^T$.
Also, should the 2nd and 3rd convolutions be labeled "8x16x1" and "16xKx1" respectively? As far as I can tell, only the first Conv1D should have a dependency on N.

Additional feedback
---------------------------
In Figure 2, the "reshape" operation should be "transpose" instead.

---

> ### Author Response · Authors · 2020-11-14
> **Clarification of problem setting and experimental protocoll**
>
> Thank you for your detailed review. We added are more explicit problem setting at the beginning of our methodology section with the hope of clarifying confusion about our definition of a task, schema, and predictors since there are a few differences from the classical problem setting in few-shot learning to focus on learning across tasks with varying schemas.
>
> During the OpenML experiments, we only split features and instances for meta-training and meta-testing since we want to show that our method can generalize to a new task that consists of unseen features. During the EMNIST experiments, there is an inherent meta-split across instances, features, and classes since these are two distinct datasets, demonstrating that by learning an alignment on EMNIST Letters, we can achieve significant improvements when sampling few-shot tasks from EMNIST Digits. Tasks are always subsets with few samples of the respective meta-dataset, and while OpenML datasets are not referred to as tasks in our paper, one can see a task as a separate dataset. We always express order- / permutation-invariance w.r.t. the schema which describes the feature space only and thus the order of features.
>
> There are no other approaches in the current meta-learning literature that deal with unstructured data with varying schemas in the first place so it is not possible to compare the results to the existing literature disregarding the metric used in our evaluation.
>
> While we understand your concern regarding a multi-modal learning paradigm, we want to clarify that our focus lies on tasks that have a varying schema but share a similar underlying distribution which enables feature alignment. We focus on addressing the issue where inputs from two different tasks but from the same domain might still differ in shape, which usually makes it infeasible to use the same model for both of them unless a common encoding or, in our case, alignment is provided. Chameleon is then, the solution to this issue.
>
> We appreciate your perspective on the similarity with attention mechanisms. We updated the related work section to include this pointer. In contrast to standard attention approaches, we specifically train the parametrized Chameleon to compute a soft permutation matrix which can realign features across tasks with varying schema when multiplied with a value matrix instead of computing a simple attention mask.
>
> In regard to your last question, yes the architecture essentially boils down to three matrix multiplications on the transposed input with nonlinearity and final softmax. The only reason we decided to use a 1D-convolution is that we have a meta-batch size during training and thus input data of form $B\times F \times N$ where $B$ is the meta-batch size, $F$ the number of features of the task, and $N$ the number of instances. Alternatively, one could reshape the 3-dimensional input $B\times F\times N$ to $BF\times N$ before using the encoder with dense layers generate the output of shape $BF\times K$ before reshaping it back to $B\times F\times K$.
>
> Regarding the design choices in general, we were focused on the basic idea and the first proof of concept. Thus, we used a relatively simple encoder which is equivariant to the feature order by simply transposing the input data. The approach can work as the first work to tackle few-shot learning across unstructured data with varying schemas, thus serving as a baseline for future research.
>
> Finally, thank you for pointing out the typo in Figure 2, we updated it to "Transpose".

---

### Official Review · AnonReviewer5 · 2020-11-05
**Confusing presentation of problem statement and method**

**Rating:** 3
**Confidence:** 4

**Review:**

Summary:
This paper aims to perform meta-learning across tasks that have different input data types by learning separate task-specific encoders, and then aligning the features produced by these encoders before making predictions.

Pros:
Sharing information across tasks with different input types is a relevant problem
Cons:
Precise problem statement and method very unclear
Experiments are only on toy datasets

Detailed Comments:

It is not clear from the abstract / introduction what is meant by “schema.” From the abstract: “for example, if the number of predictors varies across tasks, while they still share some variables.” Does this refer to the number of classes in a few-shot problem? What variables are shared? Classes, or input features? Later in the intro: “training a single model across different tasks is only feasible if all tasks share the same schema, meaning that all instances share one set of features in identical order.” These definitions of schema do not seem to be the same. Schema also does not seem to be defined in Section 3. At the beginning of that section it says, “every task has to share the same schema of common size K” which seems to indicate “schema” is the number of features and then a few lines later, “ tasks with varying input schema and feature length F” which seems to indicate “schema” is *not* the number of features.


In the related work section, few-shot learning did not begin in 2017 as might be suggested by the citations. It would be good to recognize the earlier works in this area, such as
Fei-Fei, L. et al. A bayesian approach to unsupervised one-shot learning of object categories. 2003
A Bayesian framework for concept learning. PhD thesis, Massachusetts Institute of Technology, 1999.
For few-shot learning with deep learning, Matching Networks should arguably be cited: Vinyals, Oriol, et al. Matching networks for one shot learning. 2016.
The original MAML paper actually proposed the first-order version of MAML, Nichol et al. was not the first to propose this.

I don’t understand how the method works when the features are learned and not given. For example, the encoder for EMNIST-Digits produces 32 features, while the encoder for EMNIST-Letters produces 64 features. If the meta-training tasks are drawn from only EMNIST-Digits, then how can the “re-ordering” matrix be learned from EMNIST-Digits such that it can re-order features from EMNIST-Letters? At the most basic level, based on Figure 2, the matrix \Pi would have to have different dimensionality for each dataset. Even if they were the same dimensionality, how is the feature ordering supervision performed in this case?

In the “main results”, if you sub-sample features, how do you know that the sub-sampled features have enough information to perform the classification task?

It would be helpful to have an experiment on a less-toy dataset, both to demonstrate that the problem of “mis-aligned features” exists in more complex data, and that the method can address it.

Overall, this paper is extremely confusing. I do not understand the problem statement or how the method is trained in the learned feature case. In my view, the clarity of this paper needs to be significantly improved to consider acceptance.

---

> ### Author Response · Authors · 2020-11-14
> **Clarification of problem statement and method**
>
> Thank you for your great feedback. We added are more explicit problem setting at the beginning of our methodology section, which hopefully clarifies the formalization of schemas. The schema of a task describes not only the number and order but also the semantics of the predictor variables of $X$. It does not imply any characteristic of the classes in $Y$. Please note that we denote the feature length of tasks with a varying schema with a varying $F_\tau$ for a task $\tau$ and not with a fixed $F$. Only after aligning the tasks to the same schema, they share a fixed feature length $K$. Thus, schema describes the number and order of features, but also the semantics.
>
> We did not intend to suggest that the area of few-shot learning begin in 2017, only that the work by Finn et al. (2017) was the first to introduce an optimization-based approach which finds a common initialization across tasks agnostic to model choice, which indeed sparked huge interest in the field and builds the foundation for our approach. While we already cited earlier works such as Vinyals et al. (2016) and Santoro et al. (2016) in the submitted version, we agree with your assessment and updated the related work section about few-shot learning to be more explicit about recognizing earlier progress.
>
> Chameleon is schema-invariant in that it is invariant to the order of features while it can deal with a varying number of features. This is because we fix the batch size $N$ and transpose the input of a task tau of form $N\times F_\tau$ to $F_\tau\times N$. Thus, the computed $\Pi_\tau$ of size $F_\tau\times K$ would indeed have different dimensionality in $F_\tau$ for tasks with a varying number of features in the same way a classical model can predict batches with varying size $N$. Furthermore, we sample training tasks with a subset of features in arbitrary order from EMNIST Letters, while the test tasks are sampled from EMNIST Digits. Both EMNIST datasets are embeddings of handwritten images, while the embeddings are generated with similar but different networks. The idea is that by learning to align the features from tasks sampled from EMNIST Letters, Chameleon can rapidly adapt to align similar features of EMNIST Digits during meta-testing. Note that during meta-testing, the model is still updated for three steps on the support instances of the test task, so that the alignment is also adapted w.r.t. to the final classification performance.
>
> To answer your question regarding the information content of sub-sampled features, we are mostly interested in the improvement we gain by learning an alignment across tasks with varying schemas, while the classification performance of the task compared to the whole dataset has no direct implications for our research.
>
> We agree that future work needs to encompass more complex data. However, this is the first approach to our knowledge to tackle few-shot classification for unstructured data with different schemas. Thus, we purposely wanted to introduce a simple model that can tackle the problem and show it is possible to learn across these tasks.

---

### Decision · Program_Chairs · 2021-01-07
**Final Decision**

**Decision:**

Reject

**Comment:**

After carefully going through the reviews and rebuttal, and looking at the content of the paper as well, I feel there are some issues with the current manuscript. As also pointed out by AnonReviewer5 and AnonReviewer2, the text lacks clarity. From specifically defining what a schema is, to being more explicit about the limitation of the work.
I understand that the authors are interested in a largely unexplored setting, and hence there might not be a lot of prior work to cement the evaluation protocol. Particularly because of this I think such papers need to be upfront and clear not only in what is the setting and what is the evaluation but also what are the limitations and open problems.

I do agree that there is value in this direction of research, and that the idea of re-ordering the features using attention (which I have to agree it is reminiscent of Bahdanau et al., ICLR 2015 -- though the semantics of it and its purpose makes it novel here) might be a way forward. But I do think for the paper to make an impact (and be ICLR ready) it needs more work both in the writing and maybe on the experimental side as well (consider some more complex task, or be more explicit on what is the common aspect between tasks in the distribution that can allow chameleon to work)